# Light Stress in Yeasts: Signaling and Responses in Creatures of the Night

**DOI:** 10.3390/ijms24086929

**Published:** 2023-04-08

**Authors:** Ilaria Camponeschi, Arianna Montanari, Cristina Mazzoni, Michele Maria Bianchi

**Affiliations:** Department of Biology and Biotechnology “C. Darwin”, Sapienza University of Rome, 00185 Rome, Italy; ilaria.camponeschi@uniroma1.it (I.C.); arianna.v.montanari@gmail.com (A.M.); michele.bianchi@uniroma1.it (M.M.B.)

**Keywords:** fungi, hydrogen peroxide, transcription factors

## Abstract

Living organisms on the surface biosphere are periodically yet consistently exposed to light. The adaptive or protective evolution caused by this source of energy has led to the biological systems present in a large variety of organisms, including fungi. Among fungi, yeasts have developed essential protective responses against the deleterious effects of light. Stress generated by light exposure is propagated through the synthesis of hydrogen peroxide and mediated by regulatory factors that are also involved in the response to other stressors. These have included Msn2/4, Crz1, Yap1, and Mga2, thus suggesting that light stress is a common factor in the yeast environmental response.

## 1. Introduction

Solar light is a ubiquitous energy source in the Earth’s surface biosphere. Visible light spans a very small window of the electromagnetic radiation spectrum and can be divided into different subfractions, including blue, green, and red light, depending on its wavelength. The specific subfraction of light is sensed, captured, and managed by specialized proteins and molecules that drive the light responses in organisms.

Living organisms, such as plants, algae, and bacteria, have evolved specialized photosystems managed by photons and use light in their biosynthetic pathways. However, every living organism in the surface biosphere is subjected to light, and some form of response is likely to be present in all organisms. In addition to its innate energy, light is temporally dependent because of its periodic availability associated with the Earth’s rotation. Evolution has enabled some organisms to take advantage of light and to develop useful light-dependent systems, such as photosynthesis and vision, and to develop adaptive/protective mechanisms, such as a circadian clock and associated behaviors, which minimize deleterious effects of light on cellular and organism functions. These living beings can be defined as “photophiles”. Many other organisms are incapable of exploiting light and have developed mechanisms of response and defense when subjected to it. We can classify these organisms as “skotophiles” or “lovers of the darkness” (σκότοσ). Yeasts belong to this latter group. Light is a source of stress for skotophiles, and stress response pathways in yeasts have developed a response to light, involving many different effectors and transcription factors that also support other stress responses and regulations.

In this review, we reported on the role and the effect of light in the non-photosynthetic taxonomic group of fungi. In particular, we focused on *Saccharomyces cerevisiae,* as well as other yeasts, which have been deprived of light-specific receptor systems. In the absence of any specific system that extracts energy from light (photosynthesis) or regulates biological processes based on timing of light exposure (circadian clocks), the yeast response to light appears to be oriented as a stress response in order to protect against damage, which were the main issues found in this report. For completeness, the circadian clock present in some fungi was also described, as this could correlate with certain cyclic activities in yeast.

## 2. Light as a Source of Stress

Ultraviolet light is the source of major stress in cells: UV-B (315–280 nm) causes DNA damage, such as pyrimidine dimerization [1], while both UV-A (315–400 nm) and UV-B induce the formation of intracellular ROS (reactive oxygen species), namely superoxide, hydrogen peroxide, and hydroxyl radicals. ROS can, in turn, react with DNA, protein, and lipids, generating extended cellular damage [1,2]. Visible light can also generate oxidative stress in fungi, reducing the growth rate in numerous fungal species [3]. Interestingly, the less harmful visible light (400–700 nm) has been used as a signal to promote resistance in yeast to the more stressing UV lights. This signaling pathway may depend on specific receptor systems and could include the synthesis of pigments, such as carotenoids and melanin, to absorb UV photons and/or to inactivate ROS. DNA repair enzymes, where are necessary to counteract UV DNA damage, have also been photoinduced [3].

The cellular defense against ROS is related to the synthesis of the antioxidant enzymes catalase, superoxide dismutase (SOD), and glutathione peroxidase. These enzymes are photoinduced in fungi *Neurospora* and *Aspergillus* [4,5]. The genes of the yeast *S. cerevisiae*, which may be involved in light-stress responses, are reported in Appendix A. A less obvious role in ROS resistance has been associated with heme biosynthesis, as biosynthetic intermediates have been highly photoreactive. In particular, the terminal biosynthetic enzyme ferrochelatase has been induced by light in many fungi [4,6,7]. In *S. cerevisiae*, ferrochelatase was encoded by the gene *HEM15*/*YOR176W*. This enzyme allowed the chelation of a ferrous iron cation by protoporphyrin IX and localized into the inner mitochondrial membrane [8]. A second ferrochelatase gene was found in *S. cerevisiae*, gene *MET8*/*YBR213W*, which encoded the bifunctional enzymes, dehydrogenase and ferrochelatase, involved in the final two steps of siroheme biosynthesis [9], a prosthetic group necessary for sulfate assimilation and methionine biosynthesis (sulfite reductase). The deletion of *HEM15* caused heme auxotrophy and the inability to respire, while the deletion of *MET8* generated some alteration to stress. However, neither Hem15 nor Met8 appeared to be connected to a light response (see https://www.yeastgenome.org/ for the survey, accessed on 1 March 2023). The homologous genes for both *HEM15* and *MET8* were found to be widespread among yeasts, resulting from a search for orthologous genes in the GRYC database (http://gryc.inra.fr/index.php?page=home, accessed on 8 March 2023, data reported in Appendix A).

## 3. Response to Light in the Yeast *S. cerevisiae*: Roles of the Transcription Factors Msn2 and Crz1

Stress response has been an extensively studied and well documented issue in *S. cerevisiae* [10]. It is based on signaling cascades composed of several different signaling and transcription factors that activate cellular responses to a variety of external stimuli. The transcription factors Msn2/Ymr037c [11] and Crz1/Ynl027w [12,13,14] were two components of these cascades that exerted their transcriptional activity as a consequence of cytoplasm-nucleus translocations [15]. Both were demonstrated to respond to light stress as well [16,17]. Proteins with a weak similarity to Msn2 and Crz1 have also been present in the other sequenced and annotated yeast species (http://gryc.inra.fr/index.php?page=home, accessed on 8 March 2023).

### 3.1. Msn2

Msn2 is a general transcription factor involved in stress responses [10]: In its dephosphorylated active form, it is located in the nucleus where it binds to sequences of STREs (stress response elements), which have been present in promoters of several responsive genes and modulated their transcription [18,19]. In its inactive phosphorylated form, Msn2 is located in the cytoplasm. The dephosphorylation/activation of Msn2, and its nuclear localization, were dependent on a negative correlation with protein kinase A activity (PKA) and cAMP levels [20,21].

The cellular localization of Msn2 can be mediated by light. The first studies on the light effects [16] showed that the shuttling of Msn2 between the cytoplasm and the nucleus had been induced by light. The nucleocytoplasmic oscillations of Msn2 were investigated in further studies that aimed to define the involvement of cAMP/PKA regulation [22] and the role of gene regulation [15], as well as to develop oscillation modeling [23]. More recent studies have been focused on the effect of blue light (450–490 nm) as a stressor, using Msn2 as a reporter protein. A lower light energy limit for the activation of the stress response in *S. cerevisiae* was determined [24], and a correlation between the response to light intensity and the timing of light exposure was demonstrated. The cellular localization of Msn2 has been described as the evolution of three different states: cytoplasmic localization, nuclear-cytoplasmic shuttling, and nuclear localization, corresponding to different states of the stress response. In cells exposed to continuous blue light, the evolution pattern of Msn2 localization states depended on the light intensity and the level of PKA activity and cAMP concentration [25]. An analysis at the single-cell level showed substantial heterogeneity in the Msn2 localization status and correlated the nucleocytoplasmic oscillation frequency with the blue-light-induced stress intensity and with Msn2 phosphorylation in strains with different genetic backgrounds, which then affected PKA activity and cAMP concentration.

### 3.2. Crz1

Crz1 is another transcription factor that regulates the transcription of many stress-responsive genes [26] by binding to the specific promoter sequences CDREs (calcineurin-dependent response elements) [12]. The activation of Crz1 depended on the intracellular Ca^2+^ concentration and the activity of the two conserved proteins, calmodulin and calcineurin phosphatase [27]. Ca^2+^-mediated signaling is a conserved signaling pathway [28], and it was essential when *S. cerevisiae* was exposed to several stressful conditions [27]. Similar to Msn2, nuclear translocation of Crz1 required dephosphorylation [29], and it was activated by blue light [17]. The light-induced nuclear localization of Crz1 was dependent on calcineurin and stimulated by extracellular Ca^2+^, but the dynamic of the Crz1 cellular localization followed a different pattern when Ca^2+^ was the only inducer [17].

## 4. Response to Light in the Yeast *S. cerevisiae*: Hydrogen Peroxide and Peroxiredoxin

Studies on the stress-induced transcription factors Msn2 and Crz1 have strongly indicated the unfriendly nature of light towards yeast and have classified it as a stress factor. This has led to new questions about yeast, which is an organism without a defined light sensing system. How is the light signal transduced, and what are the transducers? What are the messengers in the signaling pathway? A potential answer was supplied by investigations into the role of hydrogen peroxide and peroxiredoxin in Msn2 cellular localization after blue-light exposure [30]. In this work, a correlation between light, hydrogen peroxide, and the cellular localization of Msn2 was demonstrated. The synthesis of intracellular H_2_O_2_ was dependent on the flavin-binding peroxisomal fatty acyl-CoA oxidase Pox1/Ygl205w [31], thus associating light-sensing with fatty acid metabolism. Downstream from the blue-light-induced H_2_O_2_, the fine tuning of peroxiredoxin Tsa1/Yml28w oxidation [32] and thioredoxin (Trx1/Ylr043c and Trx2/Ygr209c) [33] oxidation cycles were shown to be critical for the nuclear accumulation of Msn2 (Figure 1). Briefly, the presence of blue light and oxygen enabled Pox1 to oxidize activated fatty acids (acyl-CoA) and to produce enoyl-CoA and hydrogen peroxide. The increased H_2_O_2_ concentration favored the oxidation of the Cys48 residue of the peroxiredoxin Tsa1 (Tsa1-SOH) and triggered Tsa1 dimerization by condensation and disulfide, as associated with Cys171 of Tsa1. The oxidized dimeric Tsa1 could then be reduced by thioredoxin (Trx) and recycled. The oxidized Trx could inactivate PKA and required NADPH/H^+^ to be regenerated into a reduced form by thioredoxin reductase (Trr). An excess of hydrogen peroxide could hyperoxidize Tsa1-SOH, which then required sulfiredoxin (Srx1) and ATP to be regenerated. Finally, the action of light on Msn2 was affected by the thioredoxin-mediated inhibition of the PKA activity and localization in the cytoplasm and the nucleus.

## 5. Response to Light in the Yeast *S. cerevisiae*: A Genome-Wide Approach

The results reported above indicated that light triggered a widespread stress response in *S. cerevisiae* and suggested that, in the absence of a proper reaction to light injuries, cellular proliferation could be severely impaired, especially in mutant strains lacking gene-encoding for proteins involved in the cellular protection against light. Following this criterion, a genomic approach has been applied with the aim to identify *S. cerevisiae* genes whose deletions have strongly affected yeast growth under visible light (400–700 nm) exposure, as compared to a wild-type strain [34]. The *YAP1/YML007W*-deleted strain, which proved to be light sensitive [35], was used as a reference strain. Tens out of hundreds of light-sensitive strains were validated. Many (Figure 2) showed correlations with oxidative stress, including H_2_O_2_-resistance and the pentose pathway that provided NADPH/H^+^ regeneration [36], as well as the HOG pathway [10]. Different stress signals, such as heat, pH, cell-wall damage, and osmolarity, activated the HOG pathway, and light could be included in this list. Interestingly, the deletion of the *MSN2* gene affected growth under visible light only when associated with *MSN4/YKL062W* deletion. Low PKA activity that enabled the nuclear localization of Msn2 was required for growth under visible light exposure. The additional genes involved in the transcription, translation, and post-translation protein activations were proven to be essential for growth under visible light conditions (Figure 2). In contrast, mitochondrial and respiratory mutants showed light resistance, as previously reported [37], and were poorly represented among the light-sensitive strains. Apparently, this finding was in contrast with the fact that light could damage cytochromes in the presence of oxygen and block respiration [38]. However, *S. cerevisiae* is a fermentative yeast with dispensable mitochondrial functions and respiration, and its physiological assets could account for this result. In summary, these genome-wide studies on the visible light response of *S. cerevisiae* have pointed to the involvement of different general cellular activities, such as general transcription, protein synthesis, and protein aggregation, and of different stress response pathways, in defense against the negative effects of light.

One crucial point of light response studies has been in the identification of a master molecular sensor that is able to transduce photons into a (bio)chemical signal. Authors have explored some possible candidates. The flavin-containing peroxisomal oxidase Pox1, which had been reported in a previous study on Msn2 localization [30], was the first choice. However, the mutant of *POX1* deletion only had weak light sensitivity during growth. The opsin and opsin-like genes, including *YRO2* and *MRH1,* as well as the stress-responsive gene *HSP30*/*YCR021C* [39] were also considered as candidates, but normal light growth in the deletion-induced mutants was observed [34]. To date, the actual nature of the photon transducer remains poorly understood. Downstream to this first step, the reduced nuclear accumulation of Msn2 and reduced glycogen synthesis appeared to be common phenotypes of the deleted strains with increased light sensitivity. Furthermore, high PKA activity was correlated with poor growth in visible light and vice versa. Yap1 could play a role in light-stress responses because visible light exposure induced the prompt nuclear localization of Yap1 and reduced the growth of a *yap1*Δ mutant strain [34]. In addition, Yap1 was responsive to hydrogen peroxide and involved in oxidative stress metabolism [10]. Considering the involvement of Msn2/4, Crz1, and Yap1 in the visible light-stress response and the spatial and temporal presence of light in the surface biosphere, we can assume that light could be the principal source of stress for yeast.

## 6. Response to Light in the Yeast *Kluyveromyces lactis*: *Kl*Mga2, a Further Light-Response Factor

The majority of yeast studies have been performed on the domesticated fermentative yeast *S. cerevisiae*. However, other yeasts have been researched, including the respiro-fermentative yeast *Kluyveromyces lactis*, which has a lactose-utilizing metabolism that is rare among yeasts, and it is used in industrial and biotechnological applications. These and other differences, as compared to *S. cerevisiae*, such as the absence of glucose repression and the indispensability of its mitochondrial functions, justified the use of this organism in basic research. The studies on the *K. lactis* transcription factor KlMga2 (ORF *KLLA0E17953g*) revealed interesting physiological interconnections with a light response. *KlMGA2* was homologous to the genes *MGA2*/*YIR033W* and *SPT23*/*YKL020C* of *S. cerevisiae*. Genes homologues, as well as weak homologues to *MGA2,* are broadly diffused among yeasts (http://gryc.inra.fr/index.php?page=home, accessed on 1 March 2023). Mga2 and Spt23 of *S. cerevisiae* were involved in fatty acid biosynthesis [40]. In particular, Mga2 was a hypoxic transcription factor involved in the transcription of the fatty acid desaturase gene *OLE1*, which was upregulated under hypoxic conditions [41]. Similarly, *Kl*Mga2 was a hypoxic regulator of lipid biosynthesis in *K. lactis* [42,43]; however, as compared to *S. cerevisiae*, it was also involved in respiration and an oxidative stress response [44,45]. The lipid content and composition were modified in *K. lactis* that had been exposed to white light: Defects in growth, mitochondrial structure, respiration rate, and hydrogen-peroxide metabolism (catalase and superoxide dismutase genes) caused by *KlMGA2* deletion were exacerbated in the mutant strain by light, suggesting an antagonistic and/or protective role for *Kl*Mga2 against light stress [46]. Interestingly, the deletion of *KlMSN2* (*KLLA0F26961g*) did not generate the same light-dependent defects/phenotypes as the deletion of *KlMGA2*. In particular, the growth, the respiration rate, and the expression of genes involved in H_2_O_2_ metabolism were not affected by light in a deletion-induced mutant. The homologue gene of *MSN4* appeared not to be present in *K. lactis* (http://gryc.inra.fr/index.php?page=home, accessed on 8 March 2023).

Light-induced ROS formation is harmful for the cell and can result in DNA damage and genetic variability. In *K. lactis*, the deletion of *KlMGA2* was pleiotropic and generated phenotypes with measurable frequency of extragenic suppression. Exposure of the deleted mutant strain to white light, darkness, or alternating white light and darkness, had no effect on the frequency of extragenic phenotypic suppression, except when the light/darkness cycle was 24 h, a condition that drastically increased the frequency of the suppression. After this occurrence, phenotypic suppression was stably maintained in the yeast population [47]. This phenomenon was a typical output of the resonance occurring between two systems oscillating at the same frequency and was indicative of the presence of a cellular mechanism in yeast with a 24 h periodicity, which responded to light/darkness cycles and was involved in the genetic arrangement of the cell. The KlMga2 protein could be involved in such a mechanism.

## 7. DNA Damage in Fungi

Exposure to UV could result in specific DNA damage. Photolyases, as a light-receptor protein, enabled the use of light energy to repair DNA damage. These proteins contained a PHR (photolyase-related region) that bound chromophores FAD and MTHF (methyltetrahydrofolate) and were widely distributed in fungi [48]. A subfamily of photolyases had diminished DNA repair capability but maintained a light signaling function, and it was named Cryptochromes and emerged from a common ancestor [49].

Photolyase was encoded by the *PHR1/YOR386W* gene (photo reactivation) in *S. cerevisiae* [50]. Photolyases were widespread among yeast species (http://gryc.inra.fr/index.php?page=home, accessed on 8 March 2023). The expression of *PHR1* was mediated by the transcription regulatory gene *RPH1*/*YER169W* [51] that, similar to *PHR1*, has been very conserved among yeasts. The Rph1 protein was a histone demethylase [52] that had also been involved in other stress-response pathways, in addition to UV light exposure [53].

## 8. Other Light Response Systems

Red light (600–850 nm) is detected by proteins named phytochromes. These proteins, highly frequent in bacteria and plants, were scarcely distributed among fungi [48], and the correlation between a red-light response and the presence of phytochromes had yet to be fully assessed.

Green light (495–570 nm) responses are mediated by rhodopsin. These are trans-membrane proteins binding the retinal as a chromophore. An animal vision system was based on the type-II rhodopsin that modulated membrane functions [54]. In bacteria and fungi, type-I rhodopsin were present and highly divergent from type II. The type-I rhodopsin were widely distributed among fungi, but studies of the correlations between these proteins and the specific light responses remain scarce. 

In *S. cerevisiae*, the putative yeast chaperone Yro2/Ybr054w had homologies with bacterial rhodopsin [55]. Yro2 is a plasma membrane protein also involved in acid stress response [56], and it was regulated at the transcriptional level by factor Haa1/Ypr008w [57]. The homologous genes of *YRO2*, and to its paralogue gene *MRH1*, have been found in other yeast species (http://gryc.inra.fr/index.php?page=home, accessed on 8 March 2023).

## 9. The Circadian Clock in *Neurospora crassa*

Exposure to light has been associated and interpreted by organisms in different ways: the presence of free space, timing, rising temperatures, and energy availability. The response followed the interpretation of the organism and depended on the presence of a specific receptor system and a downstream response mechanism. The major response to light in living organisms is photosynthesis, which converts light into biochemical energy used for biomass generation. Although fungi are not photosynthetic organisms, many fungal species have been shown to respond to light [3]. The first light system characterized in fungi was based on the photoreceptor protein White Collar-1 (WC-1) in *Neurospora crassa* [58], as it was encoded by the *wc-1* gene and responsive to the blue fraction of light (400–495 nm). The name of the gene/protein originated from the absence of carotenoid pigments in the *wc-1* mutant strains. The WC-1 protein was a Zn-finger containing transcription factors and three PAS domains (per-arnt-sim), one of which was a LOV (light-oxygen-voltage) domain and able to bind a FAD molecule [59]. WC-1, along with a second protein WC-2 [60,61], comprises the White Collar complex (WCC) and were able to bind specific LRE (light-responsive element) promoter sequences present in light-responsive genes [59,62,63]. The presence of the FAD molecule in the WCC was essential for photon absorption and the responsiveness of the system [64].

As stated above, the Earth’s rotation causes the time-dependent exposure of living organisms to light. In response, organisms have evolved circadian rhythms, which are profound biochemical/physiological changes governed by a circadian clock. A circadian clock must have a 24 h rhythm; must be independent of external stimuli, such as light; must be entrainable (capability to be reset); and compensated (resistant to external changes). In fungi, the only consolidated example of a circadian clock has been in the WCC in *N. crassa*. This clock consisted of a transcription/translation feedback loop, where the positive arm was the WCC and the negative loop was composed of a complex made by the two proteins’ FRQ (frequency) and FRH [65]. WCC regulated gene transcription by chromatin modification [66,67], and the expression of a large fraction of the *N. crassa* genome was regulated in a circadian fashion [68,69]. Therefore, WCC appeared to have a dual function: the response to the blue fraction of light and the expression of the clock-controlled genes. In addition to WC-1 and WC-2, other small proteins containing single LOV domains have been found in fungi, namely the VIVID protein [70,71] and the ENVOY protein [72]. These light-responsive proteins appeared to play a role in the fine tuning of the WCC-mediated response to light. Ascomycetes did not have WC proteins, except *Yarrowia lipolytica* [3], which had a gene (ORF *YALI0A19844g*) with weak similarities to the *wc-1* gene (http://gryc.inra.fr/index.php?page=home, accessed on 8 March 2023). This finding led to intriguing questions: Is this protein capable of sensing light? Which mechanism was involved? Are there “clocks” in *Yarrowia*? Is this organism’s evolution midway between yeasts and circadian molds?

## 10. Response to Light in the Yeast *S. cerevisiae*: Yeast Respiratory Oscillations and the Transcription Factor Yap1

Biological systems are characterized by many cyclic metabolic and physiological behaviors. The circadian rhythmicity, diffused across phylogenetic kingdoms, evolved in synchrony with light–dark alternance due to terrestrial rotation. Single-cell populations have usually been non-synchronous ergodic systems [73] and required cellular entrainment to detect their cyclic behaviors. The yeast respiratory oscillations (YROs) are metabolic cycles that were evident in yeast cells, synchronized by growth, under nutrient-limited continuous cultivation [74]. YRO was a non-circadian (1–6 h period) temperature-compensated cycle with features in common with a circadian clock [75], suggesting a possible common origin in evolution. YROs have typically been accompanied, characterized, and described by the cyclic variation of dissolved oxygen concentrations and involve, in addition to oxygen consumption, cellular division, transcription, and metabolism. Studies have proposed a possible function of YROs in mitigating the deleterious effects of oxidative stress by slowing the growth of respiring cells [76,77]. Investigations on the effects of light (white, green, or blue light, at various intensities) on YROs revealed that the period and the amplitude of cycles were affected, and the light absorption by cytochromes and respiration was likely involved [60]. Since YROs defend yeast cells from oxidative damages, the influence of the transcription factor Yap1, which regulated the transcription of oxidative stress genes [78], on YRO characteristics (period and amplitude) was investigated. However, the deletion of the *YAP1* gene had little effect beyond causing reduced growth under white light exposure. This finding indicated that other mechanisms, in addition to the light-mediated ROS production, contribute to YRO modifications.

## 11. Concluding Remarks

Light is a persistent source of energy on the surface of the Earth, and it can have harmful and stressful effects on living organisms. Light response has been directly linked to oxidative conditions and respiratory metabolism [37], as well as to hydrogen-peroxide production and signaling, suggesting possible differences in light responses when organisms have been cultivated under aerobiosis or hypoxic conditions, as well as between respiratory and fermentative organisms. For example, this has been observed between the Crabtree-negative, respiro-fermentative yeast *K. lactis* and the Crabtree-positive fermentative yeast *S. cerevisiae*, as well as between respiratory healthy cells and tissues on one side, and hypoxic tumor cells and necrotic tissues on the other side. Evolutionarily speaking, one has to consider that oxygen accumulation in the biosphere occurred much later than sunlight exposure, thus the oxidative stress evolution could be simultaneous or subsequent to the establishment of a light-stress response. Finally, given that light exposure may not be innocuous, even in the ambient environment of a laboratory, darkness cultivation appears to be a more favorable condition for yeast growth.

## Figures and Tables

**Figure 1 ijms-24-06929-f001:**
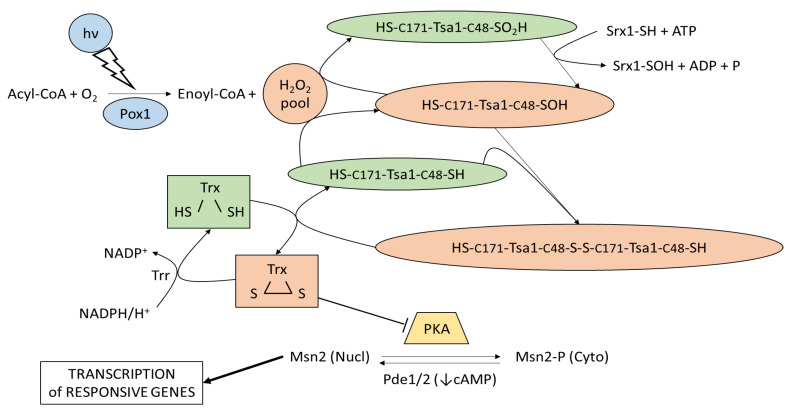
Light sensing and signaling in *S. cerevisiae*. The light-signaling cascade, from sensing up to regulation of the cellular localization of the transcription factor Msn2, modified from [30], is shown. Light (hν) is sensed by the Pox1 dependent reaction (blue) that oxidizes activated fatty acids and produces reactive hydrogen peroxide, which, in turn, triggers a cascade of sulfur-mediated oxidations (orange) involving thioredoxin peroxidase (Tsa1) and thioredoxins (Trx). Oxidized thioredoxins inhibit PKA (yellow), thus favoring the segregation of Msn2 in the nucleus and the expression of light-responsive genes. The nuclear localization of Msn2 is also favored by a low cAMP level that can be obtained by the activity of phosphodiesterases Pde1 and Pde2.

**Figure 2 ijms-24-06929-f002:**
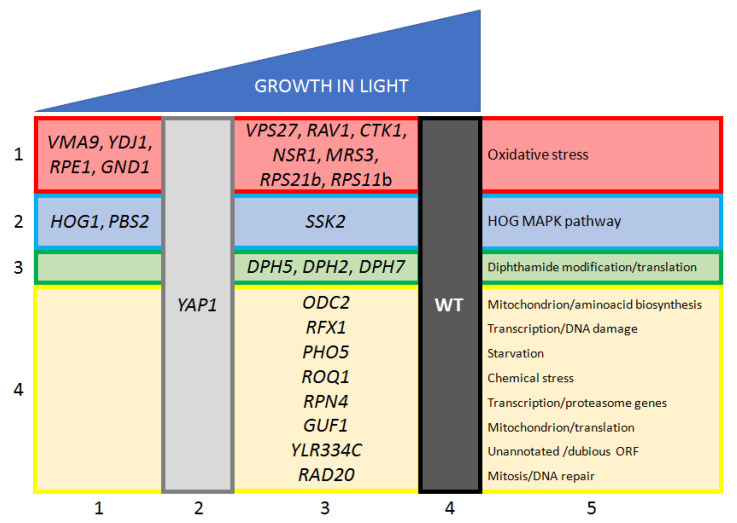
Light-sensitive mutants in *S. cerevisiae*. The graphical table reports yeast genes whose deletion proved to generate reduced growth under white light exposure (modified from [34]). Column 1 contains genes corresponding to a growth defect more severe than the defective reference strain with deleted *YAP1* (Column 2). Column 3 contains genes with a defect less severe than *YAP1* but growing significantly slower than the wild type (Column 4). Lines 1–4 group genes by function (described in Column 5): oxidative stress (line 1, red); HOG–MAPK pathway (line 2, blue); diphthamide modification (histidine modification to diphthamide in the elongation factor eEF2; line 3, green); miscellanea (line 4, yellow).

## Data Availability

Data are contained within the article or Appendix A.

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
