# Peer review of "Light Stress in Yeasts: Signaling and Responses in Creatures of the Night"

_ijms, 2023, doi:10.3390/ijms24086929_

Round 1

Reviewer 1 Report

See document

Author Response

The work of Camponeschi et al reviews the knowledge on light as source of stress in yeasts, focusing mainly on Saccharomyces cerevisiae. I consider this work of interest, but major and minor point must be taken in consideration.

MAJOR POINTS

  • In the abstract, “…and that yeast prefer darkness” must be eliminated as it is highly speculative and it is not supported by the information included. Indeed, this fact is not reviewed.

RESPONSE: the phrase has been deleted.

2- The below information on the concluding remarks must be included in the introduction. In the case to mention in the concluding remarks it must be shortened.

All these living beings can be defined as ‘photophiles’. Many other organisms remained incapable of helpfully exploiting light and, being subjected to its harmfulness, developed mechanisms of response and defense. We can classify the latter organisms as ‘skotophiles’ or lovers of the darkness (σκÏŒτοσ). Yeasts belong to this group. Light is a source of stress for skotophiles and a light stress response has been addressed to stress response pathways in yeasts, thus involving many different effectors and transcription factors operating also in other stress responses or regulations.

RESPONSE: the paragraph has been moved to the Introduction section.

3- The statement “WCC is responsive to the blue fraction of light (400-495 nm)“ is confusing.

4- The section “3. The DNA damage repair system in fungi” must be moved to other place and the title modified as it only reviews UV damage. It could be included as a subsection.

RESPONSE: the title has been changed and the section moved.

5- Some references lack and must be included after sentences:

  1. a) “Both have been demonstrated to respond also to light stress”
  2. b) “..PKA activity and cAMP concentration.”
  3. C) “…normal light growth of the deleted mutant strains was observed.”

RESPONSE: references have been added.

6- Some sentences/paragraph are very confusing as they seem to be incomplete or poorly written in English.

In cells exposed to continuous blue light, the cellular localization of Msn2 could be defined by the evolution of three different states, from cytoplasmic localization, through nuclear-cytoplasmic shuttling, to nuclear localization, corresponding to different states of stress response [9]: authors showed that the pattern of state evolution depended on light intensity and on the level of PKA activity and cAMP concentration.

Different signals, such as heat, pH and cell wall damage, activate the HOG pathway in addition to osmolarity: light can be included in the list. Interestingly, the deletion of MSN2 gene affected growth under visible light only when associated with MSN4/YKL062W deletion and low PKA activity, allowing Msn2 nuclear localization, was required for growth in this condition.

Finally, studies on the light response of S. cerevisiae, a non-photosynthetic and non-circadian organism, point to the involvement of different cellular activities, like general transcription, protein synthesis and protein aggregation, different stress response pathways, in defense of the negative effects of light.

In addition to Msn2, also Yap1 might have a role in light stress response because of the reduced growth of the mutant strain, the responsiveness to hydrogen peroxide and involvement in the oxidative stress metabolism [35], the prompt nuclear localization upon light stimulation.

The transcription factor Yap1 regulates transcription of oxidative stress genes [77], however the deletion of YAP1 gene, although causing light sensitivity, did not affect YROs suggesting that factors, other than light mediated ROS production, might cause YRO reshape.

RESPONSE: all these parts have been modified for completeness and language amelioration.

7- Section “9. Response to light in the yeast Kluyveromyces lactis: KlMga2, a further light response factor” must appear after S. cerevisiae sections.

RESPONSE: the section has been moved where suggested by the Reviewer.

8- “Mga2 and Spt23 are involved in fatty acid biosynthesis [65]”. Please, include the name of the yeast for clarification.

RESPONSE: the yeast name has been added.

9- “Interestingly, the deletion of KlMSN2 (KLLA0F26961g) did not generate the same light dependent defects/phenotypes as KlMGA2 deletion”. Please, more information must be included.

RESPONSE: detail of the defects and phenotypes have been included.

10- “Investigations on the effects of visible light on YRO revealed that period and amplitude of cycles were affected…” Explain what is affect as it is very confusing.

RESPONSE: the effect of light on YRO has been specified.

11- The section “11. Response to cycling light in K. lactis” must be eliminated as it is not a section bout “cycling light”. Accordingly, the corresponding information must appear when the KlMga2 is developed.

RESPONSE: this section has been eliminated and the content has been added to the K. lactis section.

12- The table must be organized by alphabetically ordering the genes.

RESPONSE: the table has been reorganized as suggested. Please note that the table has been moved to the supplementary material (as suggested by reviewer 2).

13- I propose a new figure including proteins acting as “stress sources”.

RESPONSE: we added a new figure (Figure 2) reporting genes involved in yeast light sensitivity. Many of these genes are actually involved in stress response.

MINOR POINTS

1- “…such as light; must…” : change “;” by “,”.

2- Rhodopsins, Thioredoxin and other proteins, in lower case.

3- “…to stresses however neither…”: change to “…to stresses. However neither…”

4- Phosphorylation/activation must be Dephosphorylation/activation

5- Cellular localization of Msn2 is mediated by light: change by Cellular localization of Msn2 can be mediated by light.

6- “…oxygen consumption: it has been proposed…”: change to “… oxygen consumption and has been proposed…”

RESPONSE: all these changes have been included in the revised text.

Reviewer 2 Report

Being a microscopist, I know that yeast may be photodamaged by excessive imaging, but I never knew that yeast is sensitive to light and it has light-protective mechanisms in place.  This review is for the people like me. As a target audience, I am interested, intrigued, and I want this review to be published. However, I also want the text to make more sense for me and thus I suggest extensive editing. Currently I am treated to a bunch of facts and genes and domains in no particular sequence, and it is hard for me to perceive inner logics in this narrative.

Here are my suggestions.

1. Please expand the introduction. Now you are discussing only generic issues. Please add the purpose of this review. State that you are interested in XYZ light effects on cells, excluding photosynthesis. Moreover, you are focusing on Saccharomyces and other yeasts, although you will discuss other fungi, if necessary for clarification of certain points.

2. Do not start with Neurospora crassa, for which practically no yeast homologs exist. This section should be at the end.

3. Tell the same story but from the different perspective, the perspective of Saccharomyces.

How about this logic: start with current section 8.

A genome-wide mutagenesis demonstrated that (surprise!) lots of genes in yeast are responsible for its growth in presence of the visible light. Discuss, which genes, pathways are involved and present a cartoon demonstrating different groups of the genes, overlap between those groups etc.

Another amazing thing -  there is a sensor pathway for sensing the light (section 7). Describe it, and  the cartoon you for this path have is OK.

This path converges on msn2? Fine. Let us talk about msn2 and crz1 (section 6). What set of the genes (what pathway?) is regulated by these TFs?

Are there other transcription factors involved? Yes, here is  K. lactis factor (section 9). What set of the genes (what pathway?) is regulated by these TFs?

After that, discuss light as a stress factor, as a ROS-generating factor, as a DNA damage factor.

Finally, light as a “pace-maker”, as a regulator of respiratory oscillations, cyclical ROS defence and, finally, possible links to circadian rhythm in Yarrowia (ending by description of the complete system in N. crassa).

4. Please, avoid the term “light” in the text. State, whenever possible “visible light”, Blue light, infrared light” etc., i.e, please be specific.

Avoid titles like “The White Collar system in N. Crassa”.  State generic issues. You are talking about circadian rhythms, so call the section “Circadian rhythms in fungi” or think of something better.  

Do not call the table 1 “S. cervisiae genes”. Give it a specific name, such as “S. cerevisiae genes which are involved in…” or think of something better. After all, those are not all 6000 yeast genes, correct?

Double-check all the sentences to make sure nothing is omitted, and they make sense. Example: “Photolyases light receptor able to use light energy to repair these DNA damages”. It feels like some verb is missing. Also, I would say “DNA damage” (singular).

I would put Table 1 into supplementary.

Try to illustrate sections graphically with cartoons whenever possible.

I recommend this review for publication after editing.

Author Response

Being a microscopist, I know that yeast may be photodamaged by excessive imaging, but I never knew that yeast is sensitive to light and it has light-protective mechanisms in place.  This review is for the people like me. As a target audience, I am interested, intrigued, and I want this review to be published. However, I also want the text to make more sense for me and thus I suggest extensive editing. Currently I am treated to a bunch of facts and genes and domains in no particular sequence, and it is hard for me to perceive inner logics in this narrative.

Here are my suggestions.

  1. Please expand the introduction. Now you are discussing only generic issues. Please add the purpose of this review. State that you are interested in XYZ light effects on cells, excluding photosynthesis. Moreover, you are focusing on Saccharomyces and other yeasts, although you will discuss other fungi, if necessary for clarification of certain points.

RESPONSE: the introduction has been expanded, as also requested by Reviewer1, and our interests and focuses have been stated.

  1. Do not start with Neurospora crassa, for which practically no yeast homologs exist. This section should be at the end.

RESPONSE: the section has been moved at the end and the title has been changed to a more general formulation (see below).

  1. Tell the same story but from the different perspective, the perspective of Saccharomyces.

RESPONSE: yeast sections have been moved up and rearranged.

How about this logic: start with current section 8.

A genome-wide mutagenesis demonstrated that (surprise!) lots of genes in yeast are responsible for its growth in presence of the visible light. Discuss, which genes, pathways are involved and present a cartoon demonstrating different groups of the genes, overlap between those groups etc.

RESPONSE: a new figure (Figure 2) has been added, showing the genes and pathways identified in the study.

Another amazing thing -  there is a sensor pathway for sensing the light (section 7). Describe it, and  the cartoon you for this path have is OK.

RESPONSE: the pathway reported in figure has been described with more detail in the text.

This path converges on msn2? Fine. Let us talk about msn2 and crz1 (section 6). What set of the genes (what pathway?) is regulated by these TFs?

Are there other transcription factors involved? Yes, here is  K. lactis factor (section 9). What set of the genes (what pathway?) is regulated by these TFs?

After that, discuss light as a stress factor, as a ROS-generating factor, as a DNA damage factor.

RESPONSE: DNA damage section and other systems section have been moved after the yeast specific sections.

Finally, light as a “pace-maker”, as a regulator of respiratory oscillations, cyclical ROS defence and, finally, possible links to circadian rhythm in Yarrowia (ending by description of the complete system in N. crassa).

RESPONSE: the section about respiratory oscillations has been moved at the end. Hypotheses about possible circadian rhythm in Yarrowia yeast have been hinted.

  1. Please, avoid the term “light” in the text. State, whenever possible “visible light”, Blue light, infrared light” etc., i.e, please be specific.

RESPONSE: the kind of light has been specified in the text when opportune. Wavelength have also been reported.

Avoid titles like “The White Collar system in N. Crassa”.  State generic issues. You are talking about circadian rhythms, so call the section “Circadian rhythms in fungi” or think of something better.

RESPONSE: title of the section has been changed as suggested.  

Do not call the table 1 “S. cervisiae genes”. Give it a specific name, such as “S. cerevisiae genes which are involved in…” or think of something better. After all, those are not all 6000 yeast genes, correct?

RESPONSE: title of the figure has been changed as suggested.

Double-check all the sentences to make sure nothing is omitted, and they make sense. Example: “Photolyases light receptor able to use light energy to repair these DNA damages”. It feels like some verb is missing. Also, I would say “DNA damage” (singular).

RESPONSE: the whole text has been checked for errors and omissions.

I would put Table 1 into supplementary.

RESPONSE: the table has been moved to Supplementary materials.

Try to illustrate sections graphically with cartoons whenever possible.

RESPONSE: a new figure has been added.

I recommend this review for publication after editing.

Round 2

Reviewer 2 Report

Thank you, I am satisfied with the revision and I recommend it for publication